# A Succession of Microbiome Communities in the Early Establishing Process of an Epilithic Algal Matrix in a Fringing Reef

**DOI:** 10.3390/microorganisms13030672

**Published:** 2025-03-17

**Authors:** Beiye Zhang, Simin Hu, Chen Zhang, Tiancheng Zhou, Tao Li, Hui Huang, Sheng Liu

**Affiliations:** 1Key Laboratory of Tropical Marine Bio-Resources and Ecology, Guangdong Provincial Key Laboratory of Applied Marine Biology, South China Sea Institute of Oceanology, Chinese Academy of Sciences, Guangzhou 510301, China; 2University of Chinese Academy of Sciences, Beijing 100049, China; 3CAS-HKUST Sanya Joint Laboratory of Marine Science Research, Key Laboratory of Tropical Marine Biotechnology of Hainan Province, Sanya Institute of Ocean Eco-Environmental Engineering, South China Sea Institute of Oceanology, Chinese Academy of Sciences, Sanya 572000, China; 4Guangdong Energy Group Science And Technology Research Institute CO., LTD., Guangzhou 510630, China; 5Sanya National Marine Ecosystem Research Station, Tropical Marine Biological Research Station in Hainan, Chinese Academy of Sciences, Sanya 572000, China

**Keywords:** microbiome community, succession, epilithic algal matrix, coral reef

## Abstract

An epilithic algal matrix (EAM) exhibits rapid expansion, recovery capacity, and high adaptability, leading to widespread distribution in degraded coral reef habitats. However, limited research on the dynamic processes of succession hinders a comprehensive understanding of EAM formation. To examine the influence of succession processes and environmental factors on the composition of EAM microbial communities, a three-factor (time × depth × attached substrate type) crossover experiment was conducted in the Luhuitou Reef Area, Sanya, China. Microbial community compositions were analyzed through 16S rRNA gene amplicon sequencing. The community was predominantly composed of proteobacteria (61.10–92.75%), cyanobacteria (2.47–23.54%), bacteroidetes (0.86–8.49%), and firmicutes (0.14–7.76%). Successional processes were found to significantly shape the EAM-associated microbial communities in the Luhuitou Reef Area. Proteobacteria played a crucial role in biofilm formation during this process, while cyanobacteria contributed to the structural complexity of microhabitats within the EAM. A chaotic aggregation stage of approximately one month was observed before transitioning into an expansion stage, eventually stabilizing into a low-diversity community. Although the relatively smooth substrate supported high biodiversity, microorganisms displayed no preference for the three different substrates. While no significant differences in community composition were observed at small-scale depths, cyanobacteria and bacteroidetes showed positive correlations with light and temperature, respectively. The EAM-associated microbial community exhibited higher complexity in the shallower regions under increased light intensity and temperature. Given the characteristics of the microbial community succession process, continuous monitoring of changes in microbial community structure and key taxa (such as proteobacteria and cyanobacteria) during EAM formation is recommended.

## 1. Introduction

Over recent decades, the epilithic algal matrix (EAM) has increasingly dominated degraded coral reef habitats [1,2]. The EAM’s intricate microstructure, formed by turf algae and microorganisms, facilitates sediment accumulation [3]. As a confluence of biotic and abiotic components, high-coverage EAM represents a significant source of primary productivity in degraded habitats [4]. The dynamic interactions between the EAM and corals have garnered substantial attention. Turf algae exhibit rapid growth rates and aggressive expansion strategies, enabling them to colonize surfaces of damaged, unhealthy, or dead corals, as well as rocky substrates [5]. However, the specific processes of EAM formation remain largely unexplored.

The characteristics of the microbial community are closely linked to the condition of the EAM. Haas et al. demonstrated a positive correlation between coral reef algal cover and microbial abundance through an investigation of 60 coral reef sites worldwide [6]. Microbial communities may play a pivotal role in EAM formation. In subtropical Western Australia, EAM re-establishes pre-cleanup conditions within 28 to 46 days post-scraping [7], in contrast to a minimum of 154 days to achieve stability on blank substrates [8]. This discrepancy is attributable to the differences between primary and secondary succession, particularly in the role of biofilms in supporting turf algae settlement during succession [9]. The EAM bacterial community is predominantly composed of proteobacteria, cyanobacteria, and bacteroidetes [10,11,12], which are also integral to coral interactions [13,14,15]. Proteobacteria contribute significantly to biofilm formation and include numerous coral-associated pathogens [16,17], as well as potential pathogens [18], which facilitate EAM expansion in the attachment matrix during coral competition. Cyanobacteria, as the primary autotrophs within this bacterial community, are crucial for maintaining the EAM’s physical structure [19] and ecological functions, such as nitrogen fixation and primary production [10,20]. Although filamentous cyanobacteria are known to drive the succession of algal turf at later stages in fringing Caribbean coral reefs [21], the bacterial community in the early stages of algal turf formation is unknown. Additionally, some studies have associated filamentous cyanobacteria (*Oscillatoria* sp.) with coral disease BBD [22,23]. Given the high relative abundance of bacteroidetes in expanding EAMs [24], the bacteroidetes to firmicutes ratio can predict competitive outcomes between coral and EAMs. Consequently, microbial community dynamics are likely closely related to EAM formation and succession.

In addition to the characteristics of the EAM, its microbial community composition is related to other biotic and abiotic factors. Most of the studies addressing the factors influencing the microbial community of EAMs have been studies that examined the EAM (the microbes in it) as a whole and its interactions with coral, or phytophagous fish [25,26,27]. In other words, in the past, more attention has been paid to the effects of biological factors, and few studies involving environmental factors have been reported. Nutrient levels, sediment loading, etc. are known to affect microbial community composition in EAMs [28,29]. Indoor experiments have demonstrated that both warming and acidification of seawater exacerbate the dominance of cyanobacteria in an EAM microbial community [30]. In addition, abundance decreases with depth due to the dependence of cyanobacteria on light [31]. However, it remains unclear how environmental factors influence differences in community composition during microbial succession in field EAMs.

With the ongoing progression of coastal urbanization, a significant portion of natural coastlines are anticipated to be supplanted by artificial substrates [32,33]. Both artificial and coral rubble substrates have been observed to support abundant EAMs in the shallow waters of the Luhuitou Reef Area [19]. Consequently, we selected a restoration substrate and an artificial construction substrate for comparison with coral rubble. Additionally, the Luhuitou Reef Area has undergone several high-temperature events [34,35] and high turbidity in the nearshore waters [36,37], making the impact of temperature and light on the characterization of EAM microbial communities a critical area of future research. This study focuses on an in situ succession experiment on EAM microbial communities, analyzing the effects of time, depth, and attachment substrates. The study aims to (1) investigate the structural succession of EAM microbial communities, and (2) assess the influence of environmental factors on community structure in the Luhuitou Reef Area. The findings of this study will enhance understanding of EAM successional processes and provide crucial information for the protection and restoration of coral reef ecosystems.

## 2. Materials and Methods

### 2.1. Research Area and the Experimental Program 

The succession experiment was conducted in the Luhuitou Reef area in Sanya (Figure 1A), where substrate units were affixed to flat reefs at depths of 1 m, 3 m, and 6 m using iron nails. Each unit was spaced a minimum of 5 m apart. The experimental unit comprised a PVC substrate and three types of attachment substrates: coral rubble (A), artificial volcanic brick (H), and cement brick (C) (Figure 1B). The surface characteristics of these substrates are detailed in Table 1. Volcanic brick (H) was selected as an optimal and cost-effective material for coral reef habitat restoration due to its structural resemblance to coral rubble (A) [38]. Cement brick (C) was employed to mimic artificial structures and to investigate its potential impact on coral reef habitat restoration.

The succession experiment was predicated on the assumption that the bacterial community in the in situ environment was in a balanced and stable state. On 11 August 2021 (day 0), in situ samples were randomly collected from depths of 1 m, 3 m, and 6 m to serve as the control group. Simultaneously, the substrate units were positioned at the predetermined depths. Subsequent samples from the substrate units were collected on days 3, 5, 7, 11, 15, 22, 44, and 100 (Figure 1B). To mitigate sample loss due to environmental conditions (typhoon interference), the number of substrate units at each depth was set to 30. Ultimately, a total of 67 samples were collected, accounting for environmental influences (Appendix A).

A HOBO^®^ Pendant device (Temperature/Light 64K Data Logger, Onset Computer Corporation, Bourne, MA, USA) was deployed at each depth to continuously monitor temperature and light variations. The surface of the HOBO was cleaned during each sampling event to maintain the photosensitive probe’s cleanliness. Continuous observations during the succession experiment indicated that temperature was affected by bottom cold-water masses at depths of 3–6 m during the early stages of succession (0–29 days) (Figure 1C). Additionally, light intensity was observed to decrease with increasing depth and over time (Figure 1D).

### 2.2. DNA Extraction and Sequencing

Biofilms and EAMs attached to the three types of substrates were collected by scraping with a sterile razor and stored in sterilized microcentrifuge tubes. The samples were homogenized using a 5 mL glass homogenizer to obtain approximately 400 mg of EAM homogenate.

Total genomic DNA was extracted from the samples following the manufacturer’s protocols. DNA concentrations were monitored using the Qubit^®^ dsDNA HS Assay Kit (Waltham, MA, USA). The preparation of next-generation sequencing libraries and Illumina sequencing was performed by Genewiz, Inc. (South Plainfield, NJ, USA). The sequencing library was constructed using a MetaVX Library Preparation Kit (GENEWIZ, Inc., South Plainfield, NJ, USA). Briefly, 20–30 ng of DNA was used to generate amplicons covering the V4 hypervariable regions of the 16S rRNA gene of the bacteria [39,40]. The forward primer sequence was ‘CCTACGGRRBGCASCAGKVRVGAAT’, and the reverse primer sequence was ‘GGACTACNVGGGTWTCTAATCC’. Indexed adapters were added to the ends of the amplicons via limited cycle PCR, and the library was purified using magnetic beads. The concentration was measured with a microplate reader (Tecan, Infinite 200 Pro, Männedorf, Switzerland) and the fragment size was verified by 1.5% agarose gel electrophoresis, with an expected size of ~600 bp. Next-generation sequencing was conducted on an Illumina Miseq/Novaseq Platform (Illumina, San Diego, CA, USA) at Genewiz, Inc. (South Plainfield, NJ, USA). Automated cluster generation and 250/300 paired-end sequencing with dual reads were performed according to the manufacturer’s instructions.

### 2.3. Community Analysis

Paired-end sequencing of positive and negative reads was performed, followed by the filtering of sequences containing ‘N’ and retaining sequences longer than 200 bp. After quality filtering and removal of chimeric sequences, the resulting sequences underwent OTU clustering using VSEARCH (version 1.9.6) with a sequence similarity threshold of 97% [41]. The 16S rRNA reference database used for clustering was Silva, version 138 [42]. Species taxonomy analysis for representative OTU sequences was conducted using the RDP classifier (Ribosomal Database Program) Bayesian algorithm [43], and community composition statistics were generated at different taxonomic levels for each sample.

Based on the OTU analysis results, the sample sequences were normalized using the vegan package in R 3.3.1 [44]. Rarefaction curves were plotted to reflect sample quality, and Venn diagrams were created to illustrate the differences in the number of shared and unique species. Alpha-diversity indices, including Richness, Shannon, Pielou, and Chao1, were calculated. Differences in alpha diversity across time points, types of attachment substrates, and depths were tested using weighted UniFrac analysis (UN). Additionally, the contribution of these three factors to variations in the alpha-diversity indices was investigated through multivariate analysis of variance (MANOVA).

The microbial community was taxonomically and functionally profiled through metagenomic sequencing. Beyond traditional phylum classifications, a literature review was conducted to categorize phyla as pathogenic, potentially pathogenic, autotrophic, probiotic, or unknown when literature information was insufficient (Appendix A). FAPROTAX (Functional Annotation of Prokaryotic Taxa) software was utilized to predict functional profiling of microbial communities using 16S rRNA marker gene sequences. The bacteroidetes to firmicutes ratio was also employed to reflect the pathogenic impact of the microbial community on coral health [24]. Indicators were logarithmized for better visualization of results. According to Roach [24], microbial communities are considered detrimental to coral health when the $\log\frac{Bacteroidetes}{Firmicutes}$ > 0.

Heat cluster analysis of the relative abundance of phylum, genus, functional taxa, and predicted functions was used to delineate succession stages. Non-metric multi-dimensional scaling (NMDS) was applied to visualize beta diversity across different stages. Clustering was assessed using ADONIS and ANOSIM statistical tests. LEfSE analysis was conducted to identify species differences between community groups. Microbial co-occurrence networks at different EAM stages were constructed to analyze community structure, including only the top 200 ASVs from the 16S dataset to reveal robust associations. Mantel tests and stepwise regression analysis were performed to determine correlations between the community and environmental factors.

Origin 2022 and SPSS 27 were employed for correlation analysis and mapping. Heat cluster maps, ADONIS and ANOSIM statistical tests, NMDS, LEfSE, and Mantel test maps were generated using Tutools (http://cloudtutu.com.cn/, accessed on 27 June 2024). FAPROTAX were generated using Tutools (http://www.cloud.biomicroclass.com/CloudPlatform, accessed on 13 August 2024). The co-occurrence network was constructed using Gephi-0.9.3 for OTUs with abundance TOP 5%, and correlations with absolute values of correlation greater than 0.4 and *p*-values less than 0.05 were screened.

## 3. Results

### 3.1. Alpha-Diversity of Microbial Communities in EAM

A total of 3,378,576 high-quality reads were obtained from the samples, resulting in the detection of 48,071 OTUs in the sequencing dataset. The rarefaction curve generated from the OTU table based on leveling and homogenization, as shown in Appendix A, suggests that the sequencing depth adequately captures species diversity of microbial communities in EAMs. Appendix A indicates a higher number of shared OTUs across different depths and attachment substrate types, as well as a greater number of unique OTUs across different time periods.

The alpha diversity of the microbial community during different succession times was assessed using Richness, Shannon, Pielou, and Chao1 indices (Figure 2). Alpha diversity exhibited significant differences throughout the succession (ANOVA: F_7,5_ > 4.876, *p* < 0.05). The highest community diversity was recorded on Day 44, with a fluctuating increase in diversity up to this point, followed by a slight decrease by Day 100. Succession time had a greater influence on alpha diversity indices compared to depth and attachment substrate type (Appendix A). No significant differences in biodiversity indices were observed between depths (ANOVA: F_2,5_ < 5.786, *p* > 0.05) (Figure 2). For attachment substrate types, differences in Richness (ANOVA: F_2,5_ = 3.647, *p* > 0.05) and Chao1 (ANOVA: F_2,5_ = 2.109, *p* > 0.05) were not significant, while significant differences were observed in Shannon (ANOVA: F_2,5_ = 14.265, *p* < 0.05) and Pielou (ANOVA: F_2,5_ = 17.734, *p* < 0.05) indices.

### 3.2. Microbial Community Compositions in EAMs

The samples were annotated and identified as consisting of bacteria and archaea, with bacteria representing up to 99.62% of the relative abundance. The microbial community composition during succession is illustrated in Figure 3. Throughout the succession, proteobacteria, including the classes alphaproteobacteria and gammaproteobacteria, was the dominant phylum. Other highly abundant bacterial phyla were cyanobacteria, bacteroidota, planctomycetota, firmicutes, verrucomicrobiota, campilobacterota, actinobacteriota, and bdellovibrionota (Figure 3A). The $\log\frac{Bacteroidetes}{Firmicutes}$ values across all samples ranged from −2.44 to 2.71 (Figure 3B). In the control group, 87.5% of the samples were primarily dominated by bacteroidetes, with 75% of the samples exhibiting an absolute ratio value greater than 1. Conversely, firmicutes dominated only one group of samples, with a low ratio of 0.19. During the succession process, bacteroidetes and firmicutes were dominant in 35% and 65% of the samples, respectively, with absolute ratio values greater than 1 in 5% and 55% of the samples, respectively.

At the genus level (Figure 3C), the most abundant genera were *Vibrio*, *Pseudomonas*, *Photobacterium*, *Psychrosphaera*, *Nesiotobacter*, *Michaud*, *Pseudoalteromonas*, *Bacillus*, and *Thalassotalea*. Within cyanobacteria, the dominant genera were *Chloroplast*, *Cyanobacteriales*, and *Synechococcales*, accounting for 67.17%, 14.98%, and 9.43% of the cyanobacterial sequences, respectively (Figure 3D). Analysis of the relative abundance of functional taxa revealed a higher presence of pathogens in the control group and a higher presence of potential pathogens and autotrophs during the succession process (Figure 3E). To explore functional changes during succession, we analyzed the microbial community functions at different stages using functional prediction. The FAPROTAX tool was employed to annotate microbial community functions, identifying 66 functional groups, with the most abundant functions being aerobic chemoheterotrophy, fermentation, nitrate reduction, and chemoheterotrophy, among others (Figure 3F).

### 3.3. EAM Microbial Communities Clustered into Three Stages

Further heatmap clustering of mean values for each time point revealed no stage divisions at the phylum level, but the abundance of proteobacteria varied significantly during succession (Figure 4A). At the genus and functional taxa levels, the succession process was divided into before and after stages (Figure 4B,D). Cyanobacteria and predicted functions emphasized differences between the control group and others (Figure 4C,E).

Combining clustering results with alpha diversity, the succession process was divided into three stages: early (days 3–22), median (days 44–100), and late (control group) (Figure 4F). Community composition varied significantly among stages (ANOSIM: R > 0, *p* < 0.01) (Table 2). No significant differences were found between depths and substrates (ANOSIM: R > 0, *p* > 0.01) (Table 2). Depth explained more community variance than substrate types in early compositions, while median stage variance was primarily attributed to depth differences (Appendix A).

Linear discriminant effect size (LEfSe) analyses at various taxonomic ranks revealed differences in succession stages (Figure 4G). Nineteen biomarkers were identified: three in the early stage, nine in the median stage, and seven in the later stage. Pseudomonas from proteobacteria was significantly more abundant in the early stage. Pseudoalteromonadaceae and rhodobacteraceae from proteobacteria, and *Chloroplast* from cyanobacteria were more abundant in the median stage. In the later stages, *Vibrio* and *Stappiaceae* from proteobacteria were the biomarkers.

### 3.4. Network Structure and Biotic Interactions in EAM

Network analysis revealed that co-occurrence patterns among microbial communities varied across succession stages (Figure 5). The correlation-based network for the early stage consisted of 197 nodes (OTUs) and 4453 edges (correlations), 186 nodes and 1084 edges for the median stage, and 110 nodes and 722 edges for the later stage (Appendix A). The decreasing number of nodes and edges indicates a transition to a simpler network over time. Negative correlations can be observed for early-stage interactions, while positive correlations predominate in the median and late stages.

Proteobacteria, cyanobacteria, and bacteroidetes were dominant throughout succession, comprising over 80% of network nodes (Figure 5A–C). Modularity analysis identified four major modules within succession networks (Figure 5D–F). The highest density was found in the early stages of the succession, indicating intensive interactions within the community at this stage (Appendix A). The median stage exhibited more modules and higher values for network diameter and average path length, while having the lowest values for average degree, average weighted degree, density, and average clustering coefficient. It indicates that in the median stage of succession, the range of interactions between species in the community is larger, but the frequency and intensity of interactions are lower. Microorganisms in the community have not yet formed distinct specific functional groups or ecological niches. The lowest density and the highest weakly connected components and average clustering coefficient in the late successional period imply that the community may have a high degree of functional redundancy and the presence of multiple subgroups with distinct ecological niche differentiation.

### 3.5. Relationship Between EAM Microbiota Composition and Environmental Parameters

Temperature during succession had a significant effect on alpha diversity (r = 0.365, *p* < 0.01) and functional taxa composition (r = 0.167, *p* < 0.05) (Figure 6). Indices such as Richness, Shannon, Pielou, and Chao1 correlated significantly with temperature (Appendix A). In community composition, bacteroidota (r = 0.750, *p* < 0.01), bdellovibrionota (r = 0.765, *p* < 0.01), verrucomicrobiota (r = 0.802, *p* < 0.01), *Photobacterium* (r = −0.556, *p* < 0.01), and *Pseudoalteromonas* (r = 0.537, *p* < 0.05) showed significant correlations with temperature.

Regarding functional taxa, probiotic (r = 0.002, *p* < 0.05), potential pathogenic taxa (r = −0.0632, *p* < 0.01), and the functions of aerobic chemoheterotrophy (r = −0.503, *p* ≤ 0.05) and chemoheterotrophy (r = −0.501, *p* < 0.06) were significantly correlated with temperature.

A significant negative correlation was found between light intensity and depth (r = −0.612, *p* < 0.01). Light intensity correlated positively with cyanobacteria (also considered autotrophs, r = 0.546, *p* < 0.05) and chloroplast function (r = 0.534, *p* < 0.05). Verrucomicrobiota showed a significant negative correlation with light intensity (r = −0.400, *p* < 0.05).

## 4. Discussion

### 4.1. Characteristics of Microbial Community Succession in EAMs

We analyzed the composition of the EAM microbial communities over a period of 100 days. Proteobacteria, a predominant bacterial phylum in EAMs, maintained a high abundance (61.10–92.75%) but underwent significant community succession. To visualize this process, we divided it into three stages: early, median, and later stages.

Species turnover in marine benthic communities generally begins with bacterial aggregation and biofilm formation [45,46,47]. The early stage lasted approximately one month, with stochastic processes allowing organisms from the open ocean to compete for community formation. Limited sediment accumulation facilitated the dominance of *Pseudomonas* (52.17%), which thrives in nutrient-poor conditions through chemoheterotrophy. Cyanobacteria exhibited relatively high abundance (12.13%), supporting previous findings by Zhou et al. [19]. The bacteroidetes to firmicutes ratio indicated uncertainty regarding pathogenic risk in the early stage.

As succession proceeded, sediment accumulation supported species diversification, enabling nitrate reduction and fermentation functions, fostering interconnections [48,49,50]. This led to a more diverse and modular bacterial community in the median stage, with a shift in dominant microorganisms from *Pseudomonas* to *Vibrio* (25.62%). Rhodobacteraceae emerged as a biomarker, indicating ongoing biofilm formation. Cyanobacterial composition changed, with picocyanobacterial *Synechococcales* replaced by filamentous *Phormidiaceae*, *Nostocaceae*, and macroalgal epiphytic *Xenococcaceae*. The bacteroidetes to firmicutes ratio peaked at 58.63. By day 100, the microbial community had not yet reached a steady state. Based on Carvalho et al.’s study, the median stage may last 3–4 months [8]. Long-term observation of the EAM succession process in Luhuitou is recommended to elucidate the definitive succession length and dynamics of community composition. Compared to the early stage, the median stage supports a more three-dimensional microhabitat structure, refined functions, and poses a higher threat to coral health.

In the late stage, species composition simplification [50] and enhanced interactions led to a mature and stable community. The predictive function manifested in the homogenization of four main functions. Vibrio showed significantly higher abundance (67.49%), while cyanobacteria and the bacteroidetes to firmicutes ratio declined to 2.47% and 0.32, respectively. These structural and functional characteristics indicate a balanced and stable community, reducing the threat to coral health from the EAM bacterial community.

### 4.2. Effects of Attached Substrate Type on Microbial Composition

Studies on surface roughness suggest rougher surfaces are more conducive to bacterial attachment [51]. Cement brick (C) had a smoother surface due to fewer holes and smaller hole areas compared to coral rubble (A) and artificial volcanic brick (H). Paradoxically, the alpha-diversity of the microbial community was higher on C than on A and H. This may be due to dominance on rough surfaces facilitating rapid colonization, resulting in lower overall abundance. Smoother substrates may reduce the dominance of easily attachable bacteria, fostering a more diverse community. However, substrate differences (either between artificial and natural substrates, or differences in substrate surface roughness) had no significant effect on microbial community composition (Table 2, Appendix A). Simplified measurements of substrate surface characteristics might obscure the relationship between substrates and microbial community composition [51,52]. Selecting more suitable substrate-related parameters in future studies is recommended to elucidate differences in microbial community structure during early EAM stages. Overall, the results of similar microbial community composition and higher α-diversity of artificial substrates than natural substrates suggest that an increase in offshore artificial substrates may support the expansion of EAMs.

### 4.3. Effects of Depth on Microbial Composition

Environmental factors such as light intensity and temperature decrease with increasing depth, influencing microbial community trends. Coral diseases are most prevalent at elevated temperatures, with pathogen interactions being complex and influenced by environmental factors [14,16,17,18,22,23,53,54,55,56,57]. Elevated temperatures increase microbial community abundance and metabolic capacity [58]. This study found that temperature differences at different depths hardly alter the dominance of potential coral-associated pathogens and pathogenic bacteria in the EAMs of Luhuitou Reef area. Adequate light intensity supports cyanobacterial growth and reproduction [20]. Cyanobacteria and chloroplast functions were more abundant in shallow waters, indicating more structurally and functionally complex EAM-associated microbial communities. Identifying specific genera or species of most cyanobacteria remains challenging, making it difficult to elucidate differences at varying depths. High temperatures and light intensities promote complexity in cyanobacteria and functional taxa composition, suggesting future research should focus on EAM-associated microbial community dynamics during extreme high-temperature events to inform shallow-sea coral reef protection.

## 5. Conclusions

This study investigated the dynamics and factors influencing microbial community composition in EAMs by observing the succession process with three types of attached substrates at different depths. The EAM-associated microbial communities underwent a series of stages (aggregation–expansion–stabilization). Microorganisms did not selectively attach to the three substrates. While there were no significant differences in community composition at small-scale depths, communities formed in shallow waters exhibited greater complexity. Although changes in environmental factors in shallow-water coral reefs were not shown to have a significant effect on EAM microbial community characteristics in this study, the importance of these factors should not be overlooked in practice at larger spatial scales due to cumulative effects. Given the complex functions of EAM-associated microorganisms in the Luhuitou Reef area, studying the succession of their community composition is crucial for coral reef conservation. Therefore, we recommend that future studies of these factors be supplemented with consideration of their potential long-term and indirect effects in order to fully explore their impacts on EAM microbial communities.

## Figures and Tables

**Figure 1 microorganisms-13-00672-f001:**
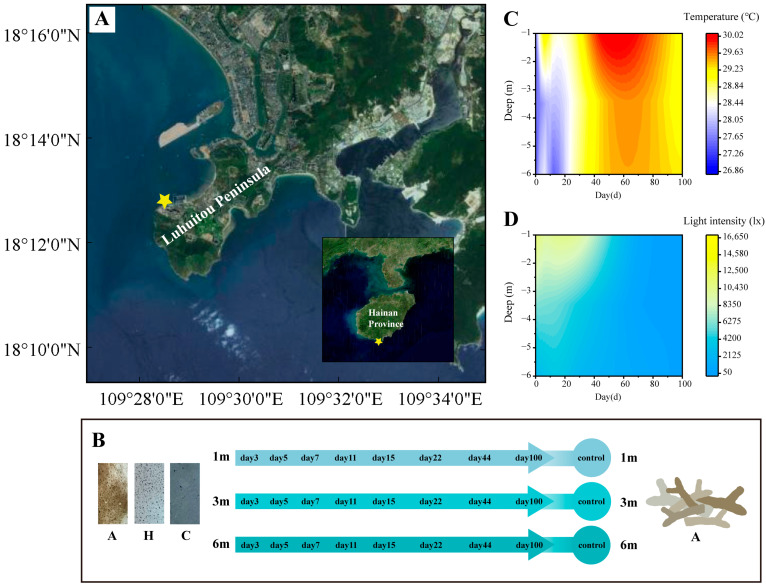
Design and environmental parameters of succession experiment: (**A**) study sites on Hainan Island; (**B**) design of succession experiment and substrate units; (**C**) temperature during succession experiment; (**D**) light intensity during succession experiment.

**Figure 2 microorganisms-13-00672-f002:**
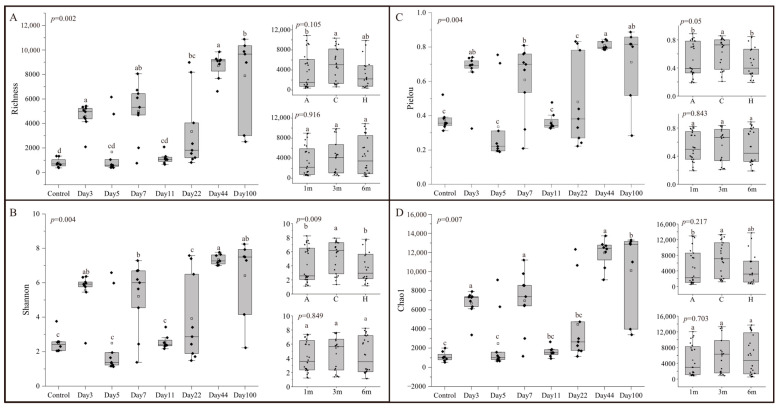
Alpha-diversity indices of microbial communities for different times, depths, and types: (**A**) Richness; (**B**) Shannon; (**C**) Pielou; (**D**) Chao1. For all variables with the same letter, the difference between the means is not statistically significant. If two variables have different letters, they are significantly different.

**Figure 3 microorganisms-13-00672-f003:**
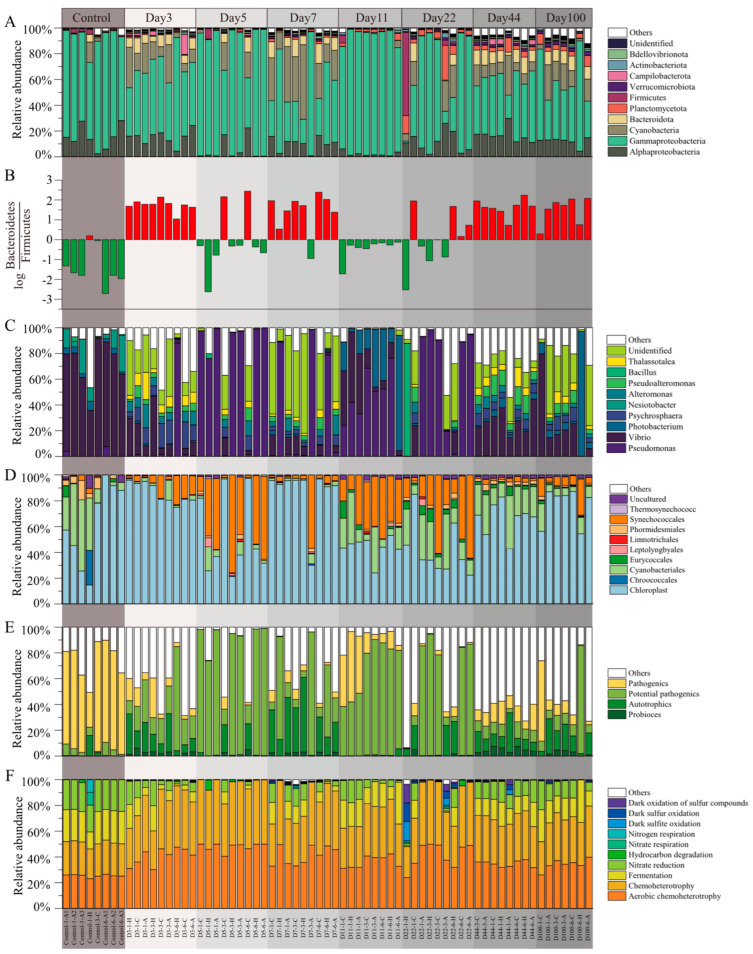
Composition characterization of microbial communities in EAM succession. (**A**) Relative abundance of phylum (proteobacteria was expressed in terms of outlines as categorical units); (**B**) $\log\frac{Bacteroidetes}{Firmicutes}$; (**C**) relative abundance of genus; (**D**) relative abundance of genus from cyanobacteria; (**E**) relative abundance of functional taxa, and (**F**) relative abundance of predicted function.

**Figure 4 microorganisms-13-00672-f004:**
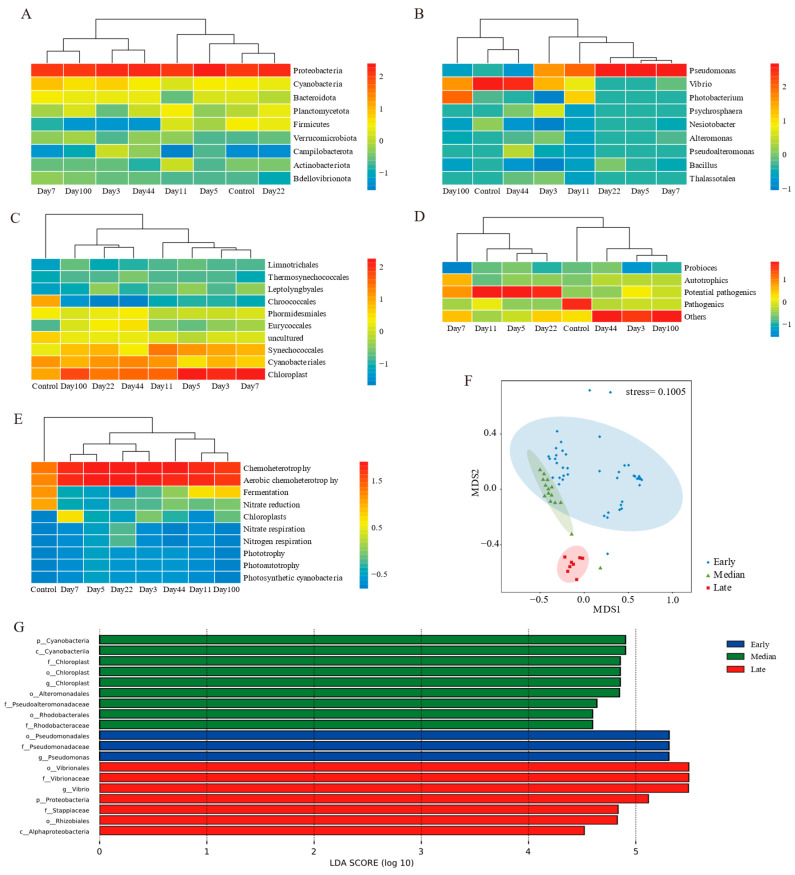
Microbial communities clustered into three stages. Heatmap clustering of (**A**) phylum, (**B**) genes, (**C**) genes from cyanobacteria, (**D**) functional taxa, and (**E**) predicted function level; (**F**) nMDS analysis of microbial communities in EAM succession; (**G**) LDA effect size cladograms comparing different succession stages of microbial community profiles.

**Figure 5 microorganisms-13-00672-f005:**
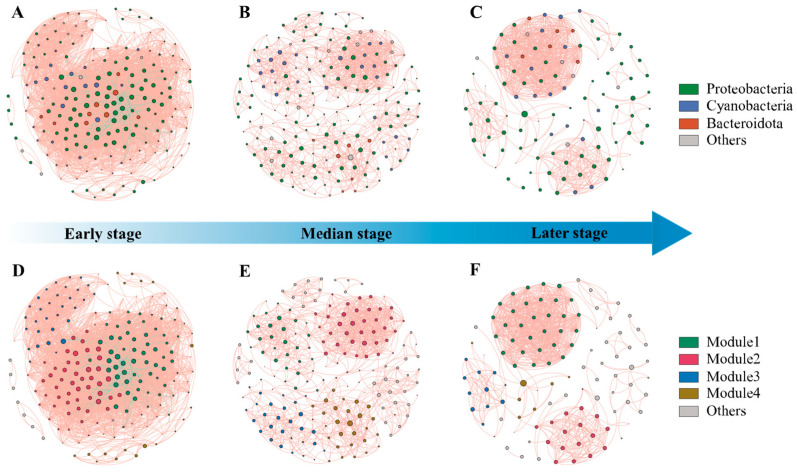
The co-occurrence networks of bacterial taxa in three stage s of EAMs. Positive and negative associations are represented as red and green lines, respectively. (**A**) co-occurrence networks of phylum in early stages; (**B**) co-occurrence networks of phylum in median stages; (**C**) co-occurrence networks of phylum in late stages; (**D**) co-occurrence networks of module in early stages; (**E**) co-occurrence networks of module in median stages; (**F**) co-occurrence networks of module in late stages.

**Figure 6 microorganisms-13-00672-f006:**
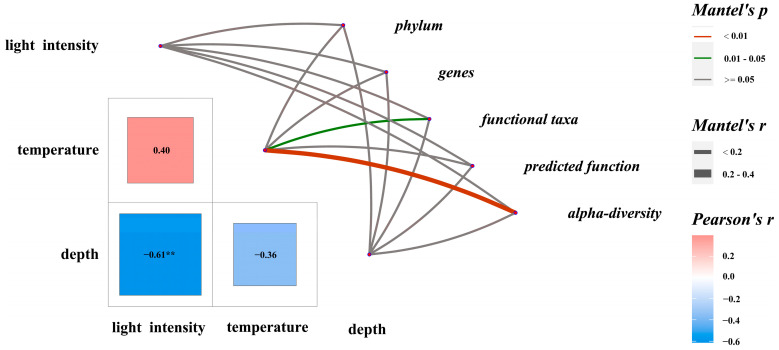
Mantel tests on characteristics of major biological groups in EAMs and potential influencing factors. ** indicates a significant difference between environmental parameters.

**Table 1 microorganisms-13-00672-t001:** Surface characterization of three attachment substrates.

Attachment Substrate Types	Number of Holes (ind./mm^2^)	Average Area of Holes (mm^2^)	Proportion of Hole Area
Coral rubble (A)	0.19	0.42	7.98%
Artificial volcanic brick (H)	0.60	0.14	8.40%
Cement brick (C)	0.04	0.52	2.08%

**Table 2 microorganisms-13-00672-t002:** ANOSIM test on stage, depth, and type in microbial community OTU. The sig column labeled *** indicates a significant difference between groups.

Variable	Group	R	*p*_Value	Sig
stages	later/early	0.479	0.001	***
	later/median	0.860	0.001	***
	early/median	0.175	0.001	***
depth	1 m/3 m	0.016	0.252	
	1 m/6 m	0.029	0.143	
	3 m/6 m	−0.028	0.787	
type	A/H	−0.008	0.532	
	A/C	−0.016	0.666	
	H/C	0.0552	0.100	

## Data Availability

The datasets presented in this study can be found in online repositories. The names of the repository/repositories and accession number(s) can be found below: https://www.ncbi.nlm.nih.gov/; the DNA sequencing data can be found in the National Center for Biotechnology Information (NCBI) under the BioProject accession number PRJNA1148297.

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
