# Peer review of "A Succession of Microbiome Communities in the Early Establishing Process of an Epilithic Algal Matrix in a Fringing Reef"

_microorganisms, 2025, doi:10.3390/microorganisms13030672_

Round 1
Reviewer 1 Report
Comments and Suggestions for Authors
This study examined microbial succession in the epilithic algal matrix (EAM) at Luhuitou Reef, China, using a three-factor experiment and 16S rRNA sequencing. Proteobacteria dominated biofilm formation, while Cyanobacteria contributed to habitat complexity. A chaotic aggregation phase (~1 month) preceded expansion and stabilization into a low-diversity community. Light and temperature influenced Cyanobacteria and Bacteroidetes, with shallower areas showing greater microbial complexity.
Line 61-67, The discussion on Cyanobacteria is overly lengthy. Please summarize the key points concisely and remove redundancy to improve clarity and readability
Line 75 - 82, Please expand this section into a separate paragraph. Discuss the factors that influence EAM microbial communities, such as depth, temperature, and other environmental conditions. Additionally, consider incorporating relevant studies to support and enrich the discussion
Line 73, This sentence does not clearly convey your overall research gap and motivation. Please revise the introduction to explicitly address: 1) Why pathogenic risks on EAM microbial communities are important, and 2) Why temperature and depth are key factors. Clearly state the research gap to strengthen the rationale for your study.
Line 106, I don’t understand this statement: 'To mitigate sample loss due to environmental conditions, the number of substrate units at each depth was set to 30.' Why specifically 30? Please clarify the rationale behind this choice.
Line 110, Is HOBO an abbreviation? If yes, please add the full name.
Line 152, Please brief describe the random sampling methods here.
Line 181, Please describe the parameters used to build the network in more detail. Specify the correlation method applied and the threshold values for r and p.
Line 204, Please correct Figure 2C to ensure it follows a same format from the other three figures.
Line 235, Change the color of Figure 3A to distinguish different samples
Line 263, How to explain the discrete points in the middle stage (green) in the MDS graph.
Line 284, Is it possible to distinguish positive and negative correlations in your network visualization? Additionally, please describe the method used to construct the network. If feasible, adjusting the node size based on the node degree would improve clarity.
Line 343, What is the dominate species on the rough surfaces in your experiments?
Line 352, Effects of deep or depth? on Microbial Composition
Line 379,In the conclusion, it would be helpful to add more environmental implications rather than a repetitive statement of the results.
Comments on the Quality of English LanguageStill need to be improved.
Reviewer 2 Report
Comments and Suggestions for Authors
Reviewers comments on article Succession of microbiome communities in the early establishing process of epilithic algal matrix in a fringing reef submitted to Microorganisms MDPI Journal
General comments:
Line 169: log(Bacteroidetes-to-Firmicutes ratio) – please use mathematical annotation
Line 225: Chloroplast genus in Cyanobacteria?
Line 236-240: Very strange font sizes and formats in the text
Figure 5. Can you please explain to me the scientific significance of this type of graphs? Nodes and connections between different nodes and differentiation by color looks very nice but I could not find the significance of it.
After presenting graphically very large data in Results, comes very short, plain and inconclusive Discussion section. If You are confident in the data that are presented in various modern graphical solutions please provide extensive discussion that will compare your data to the literature. At the last few months I found at least four similar articles that use similar methodology and graphical solutions. Please compare your findings with them.
The same applies to conclusions section.
Reviewer 3 Report
Comments and Suggestions for Authors
The authors present a fascinating document that explores the succession of microbiome communities in reef communities. I would like to ask the authors to improve the document by answering the following questions
Given that the authors have identified three distinct phases (early, intermediate, and around late) in the succession of the microbial community, how do you expect these stages to evolve if the observation period is extended beyond 100 days?
Considering that your results indicate that temperature and light intensity significantly influence microbial structure, in what way do you think climate change—particularly the increase in ocean temperatures—will affect the succession and long-term stability of the EAM (epilithic algal matrix)?
Even with differences in surface properties, the microorganisms were not strongly biased toward any single substrate type. Do you think this finding would be replicated in other reef environments, or could site-specific factors play a role in the substrate selection of T. lemma?
The study also reveals that the microbial communities, especially during the intermediate phase, cause a potential pathogenic risk. Do you detect direct relations between these pathogenic bacteria and coral health in the Luhuitou Reef Area?
You used the Bacteroidetes-to-Firmicutes ratio as a marker of microbial pathogenicity. How does this parameter relate to other established indicators of microbial health in coral reef ecosystems?
While the study suggests that cyanobacteria induce structural complexity in the EAM succession, its role in succession has been debated. Can you tell us a little more about this: If cyanobacteria in this context help or hinder recovery of the coral reef?
Given the increasing use of artificial substrates in reef restoration projects, how do you think the microbial communities that will develop on these artificial substrates differ in the long term from those found on natural coral rubble?
Because the overall network analysis reveals a decrease in the complexity of microbial interactions over time, what do you think are the ecological consequences of this simplification for the reef ecosystem?”
Considering that the study was conducted over 100 days, do you think an extended observation period (one year or more) might reveal additional dynamics in microbial succession?
The functional profiles of the microorganisms are shown to vary throughout the succession stages. How do you anticipate these functional changes will influence coral reef resilience and future restoration strategies?
Round 2
Reviewer 2 Report
Comments and Suggestions for Authors
No further questions.